# Exploratory Analysis of COVID-19 Related Tweets in North America to Inform Public Health Institutes

**Hyeju Jang[1, 2], Emily Rempel[2], Giuseppe Carenini[1], Naveed Janjua[2]**
[1]Department of Computer Science, University of British Columbia
[2]British Columbia Centre for Disease Control
{hyejuj, carenini}@cs.ubc.ca
{emily.rempel, naveed.janjua}@bccdc.ca

## Abstract

Social media is a rich source where we can learn about people's reactions to social issues. As COVID-19 has significantly impacted on people's lives, it is essential to capture how people react to public health interventions and understand their concerns. In this paper, we aim to investigate people's reactions and concerns about COVID-19 in North America, especially focusing on Canada. We analyze COVID-19 related tweets using topic modeling and aspect-based sentiment analysis, and interpret the results with public health experts. We compare timeline of topics discussed with timing of implementation of public health interventions for COVID-19. We also examine people's sentiment about COVID-19 related issues. We discuss how the results can be helpful for public health agencies when designing a policy for new interventions. Our work shows how Natural Language Processing (NLP) techniques could be applied to public health questions with domain expert involvement.

## 1 Introduction

Globally, more than 10 million people have been diagnosed with COVID-19 infection and 500,000 people have died as of June 29, 2020. Since there is no vaccine or effective treatment, governments across the world implemented wide-ranging non-pharmaceutical interventions such as hand hygiene, face masks, contact tracing, isolation and quarantine, and physical (social) distancing through banning mass gatherings and lockdowns to reduce the transmission of SARS-CoV-2.

Similar to previous infectious disease outbreaks such as Ebola, people are using social media to share news, information, opinions and emotions. Thus, social media is an important source to learn about people's reactions and concerns, and this information can be beneficial for public health institutes when designing interventions. Understanding both people's reactions to COVID-19 and where their concerns lie helps to tailor public health strategy and ultimately create better informed interventions.

In this paper, we investigate Twitter user's reactions to COVID-19 in North America, especially focusing on Canada. We analyze COVID-19 related tweets using topic modeling and Aspect-Based Sentiment Analysis (ABSA) using human-in-the-loop, and interpret the results with public health experts. We first discover topics in COVID-19 related tweets using a widely used topic modeling approach, Latent Dirichlet Allocation (LDA) (Blei et al., 2003). To assess changes in topics of discussion over time, we compare timelines of topic distributions and timing of implementation of public health interventions for COVID-19. Furthermore, we examine the sentiment of tweets about COVID-19 related aspects such as social distancing and masks, by using ABSA based on domain specific aspect and opinion terms. We also discuss how the results can inform public health interventions. Our work shows how Natural Language Processing (NLP) techniques could be applied to public health questions with domain expert involvement.

## 2 Related Work

Topic modeling and sentiment analysis have been widely used to identify issues and people's opinions in public health. In this paper, we will only introduce a few examples relevant to COVID-19. Liu et al. (2020a) applied topic modeling to identify patterns of media-directed health communications and discuss the role of the media during the COVID-19 pandemic in China. Sha et al. (2020) also identified topics related to COVID-19 on tweets posted by U.S. governors and presidential cabinet members. Sharma et al. (2020) used topic modeling and sentiment analysis to provide topic and trend analysis as

| # | Representative words | Interpretation |
|---|---|---|
| T1 | social, distancing, outside, park, walk, quarantinelife, stayhome | Social and physical distancing, including spending time outside during quarantine. |
| T2 | china, travel, canada, russia, flights, trade, border | Air travel and regional border restrictions/outbreaks. |
| T3 | hands, wash, health, public, use, need, safety | Hand washing and what people can do to prevent COVID-19. |
| T4 | home, stay, safe, work, sick, family, essential | The need to stay home and the impact of COVID-19 on essential workers and family. |
| T5 | positive, testing, tested, cases, patients, hospital, data | This topic focuses on data, particularly number of tests and cases. |
| T6 | masks, wear, face, hand, sanitizer, gloves, surgical | Things we can do to prevent COVID-19, e.g., masks and face coverings. |

Table 1: Topics most relevant to public health and their interpretations.

well as public sentiment towards social distancing and work-from-home policy interventions.

Our work is distinct from other efforts in that public health experts are actively involved in the process, with the specific goal of informing public health interventions. Our results are interpreted by these public health experts, and we use a human-in-the-loop ABSA approach to obtain domain specific aspect and opinion terms.

## 3 Data

We use a public twitter dataset about the COVID-19 pandemic, collected by Chen et al. (2020) using numerous COVID-19 related keywords such as "coronavirus", "COVID-19" and "pandemic". The data collection started on January 28, 2020, and is still ongoing, which has published over 123 million tweets.

For our work, we select tweets from January to May, whose location is Canada and United States (US).[1] Among the 372,711 tweets in total (Canada: 30,235, US: 342,476), we only include tweets written in English using tweet metadata and the spacy-langdetect toolkit[2]. This process resulted in 319,524 tweets in total, 25,595 for Canada, and 293,929 for US. To remove tweet specific keywords and urls, we use the tweet-preprocessor toolkit[3]. We do not remove hashtags and mentions because they can be informative for our work. We lower-case, tokenize using the Spacy toolkit (Explosion, 2017). Since the methods we use in this paper are all unsupervised, we do not split the data for

training and test.

## 4 What Do Twitter Users Discuss about COVID-19?

To discover topics and track the topic change over time, we construct topic models on our Twitter data using LDA[4] implement in the scikit-learn package (Pedregosa et al., 2011). We choose a model with 20 topics among 5, 10, 20, and 50 because 20 topics showed diverse and less redundant topics when manually examined.

The discovered topics are highly related to public health promotions and interventions, such as physical distancing, border restrictions, hand washing, staying-home, and face coverings, as shown in Table 1. Other topics include US President Donald Trump, initial outbreaks in Wuhan, economic concerns and negative reactions. The entire set of topics is listed in Appendix.

The most prevalent topics in Canada and US show differences, as shown in Table 2. In both countries, age and COVID-19 transmission is the most prevalent topic. The discussion around the initial outbreak in Wuhan and US President Trump's statement was also active in both countries. However, the topic about air travel and regional border restrictions is highly ranked only in Canada whereas the topic is not even listed in the top-10 in the US. Similarly, the topics about COVID-19 being like the flu and staying home are highly ranked in the US tweets but ranked lower than other topics in the Canadian tweets.

---

[1]Note that tweets with location tags are limited.
[2]https://pypi.org/project/spacy-langdetect/
[3]https://pypi.org/project/tweet-preprocessor/

[4]We also built a Non-negative Matrix Factorization (NMF) model, but LDA results were more distinct in categories according to public health experts' assessment.

| # | Canada | United States |
|---|--------|---------------|
| 1 | Age and COVID-19 transmission, as well as time. | Age and COVID-19 transmission, as well as time. |
| 2 | Initial outbreak in Wuhan. | US President Trump's statement. |
| 3 | US President Trump's statement. | Early debate on whether coronavirus is like the flu. |
| 4 | Thank yous related to the pandemic mixed with discussion of cruise ship outbreaks. | Initial outbreak in Wuhan. |
| 5 | Air travel and regional border restrictions/outbreaks. | The need to stay home and the impact of COVID-19 on essential workers and family. |

Table 2: Top-5 prevalent topics in Canada and United States.

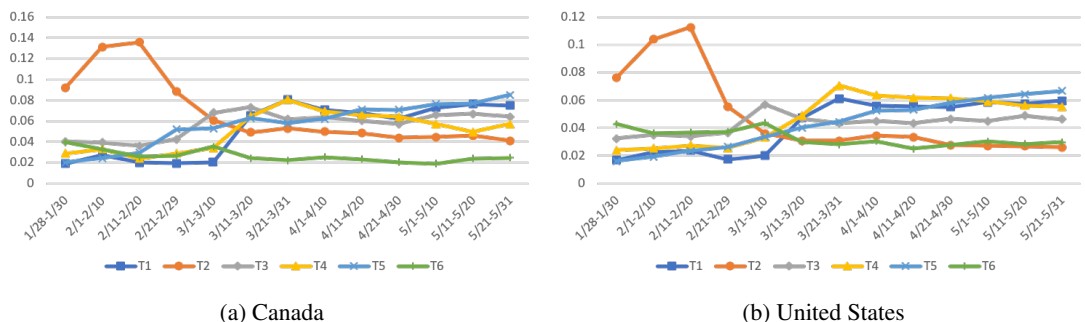

(a) Canada                    (b) United States

Figure 1: Topic changes over time. T1: social and physical distancing, T2: air travel and regional border restrictions/outbreaks, T3: hand washing and preventive measures, T4: the need to stay home and impact of COVID-19 on essential workers and family, T5: number of tests and cases, T6: masks and face coverings.

To analyze the dynamics of public health relevant topics, we investigate the change in the prevalence of the topics over time. We present a basic analysis based on examination of the estimates of $\theta$, a document-to-topic distribution, produced by the model. We first divide tweets into 10-days time buckets, e.g., Jan. 21 – Jan. 31, Feb. 1 – Feb. 10, and Feb. 11 – Feb. 20. Note that we use time in UTC-12 because tweet timestamps are in the time zone. Then, we compute a mean $\theta$ vector for tweets in each bucket as in (Griffiths and Steyvers, 2004). Based on the mean $\theta$ vector for each bucket, we draw graphs of public health relevant topics over time as shown in Figure 1.

First, we can observe that the patterns in the US tweets and Canadian tweets are very similar. Although there are slight differences, the overall increase and decrease patterns are almost identical. For example, the topic about air travel and regional border restrictions (T2) shows a peak in February and drastically decreases. Second, we can see that the topic trend is highly related to public health interventions. For example, the topic about social

distancing (T1) starts to increase in early March after COVID-19 became an issue in Seattle, US, at the end of February. Hand washing (T3) also started to be emphasized then. The topic about the need to stay home (T4) starts to increase around the end of March. In Canada, the Federal Quarantine Order was issued on March 24, and in the US, many states issued stay-at-home orders around that time as well. Discussion about the number of tests and cases (T5) gradually increases. Interestingly, the topic about masks and face coverings (T6) slightly decreases from March this is possibly because public health institutes in both countries announced their position about masks around that time.

## 5    How Do Twitter Users React to COVID-19?

Understanding people's reactions to the COVID-19 pandemic is important to public health agencies because it informs how public health agencies should frame their health messaging. To capture

sentiment revealed in tweets towards important aspects of COVID-19, we use ABSA. In our work, aspects can include public health interventions or issues associated with COVID-19 such as "social-distancing", "reopening", and "masks". We investigate people's opinion (positive/negative) towards these aspects.

We use ABSApp, a weakly-supervised ABSA system (Pereg et al., 2019). We choose ABSApp because it does not require labeled data for training, and allows manually editing domain-specific aspect and opinion lexicons produced by the method. This feature is particularly beneficial for us because we can select/add aspects public health agencies are interested in.

After training the tweet data using ABSApp, we obtained 806 aspect terms and 211 opinion terms. Editing the lexicons by public health experts resulted in 545 aspect terms (e.g., "vaccines", "economy", and "masks") and 60 domain specific opinion terms (e.g., "infectious"- negative, and "professional"- positive). Then, these manually edited terms were used for inference of sentiments.

The results of selected aspects are shown in Figure 2. We observe that the sentiment about the coronavirus outbreak itself is dominantly negative. With this, the twitter users' reactions to preparedness and misinformation appear to be more negative than being positive, suggesting the frustration about the current situation and misinformation. The mixed sentiment about masks might reflect the conflicting situation around mask usage. The negative sentiment towards asians found may imply the Anti-Asian social phenomenon escalated due to COVID-19.

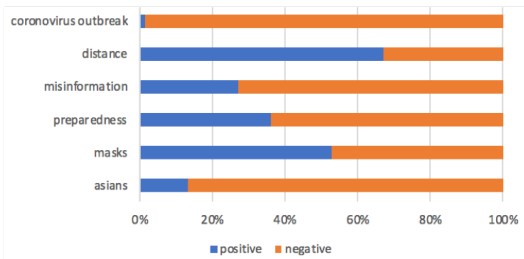

Figure 2: Aspect-based sentiment analysis results for selected aspects. The ratio between # of positive occurrences and # of negative occurrences.

## 6 Discussion

The topics identified demonstrate Twitter users' focus on discussing and reacting to public health interventions. This kind of information helps public health agencies to understand public concerns as well as what public health messages are resonating in our populations who use Twitter. For example, public health agencies in North America have focused their messaging around encouraging hand hygiene, limiting physical contact when sick and staying home to prevent infection. We can see this messaging echoing in the topics around hand washing, staying home, mask-wearing and social/physical distancing.

The change over time in the age topic shows an area for improving our public messaging. Early public health data showed a strong correlation between age and severe COVID-19 outcomes (Liu et al., 2020b; Nikolich-Zugich et al., 2020). However, we now know that SARS-CoV-2 transmission does not show the same correlation (Ontario Agency for Health Protection and Promotion, 2020). A renewed focus on messaging to highlight this difference would likely benefit our public health response.

ABSA has the potential to track stigma related to COVID-19. Our communities of Asian ethnicity have experienced unprecedented stigma and discrimination due to COVID-19. If we can understand the nature and change in stigma over time, we can develop counter-acting messages and measures.

## 7 Conclusion

In this paper we presented the exploratory results of topic modeling and ABSA on COVID-19 related tweets in North America. We compared topic modeling results of Canada and the US, and also showed public health intervention related topic changes over time. In addition, our human-in-the-loop aspect-based sentiment analysis showed twitter user's reactions about COVID-19 related aspects, which can be beneficial for public health policy makers. The limitation of this work is that we used only a small set of Twitter data because tweets with the location information were limited compared to the whole data set. However, this study is still useful for public health interventions and messaging. For future work, we will analyze our results with other types of public health data such as patient statistics for different regions or mobility information.

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

# A Appendices

## A.1 Topics discovered by topic modeling

| #   | Representative words | Interpretation |
| --- | --- | --- |
| T1  | social, distancing, outside, park, walk, quarantinelife, stayhome | Social and physical distancing, including spending time outside during quarantine. |
| T2  | corona, beer, stupid, die, cure, flu, drink, cold | Early debate on whether coronavirus is like the flu and around corona beer sales. |
| T3  | china, travel, canada, russia, flights, trade, border | Air travel and regional border restrictions/outbreaks. |
| T4  | hands, wash, health, public, use, need, safety | Hand washing and what people can do to prevent COVID-19. |
| T5  | home, stay, safe, work, sick, family, essential | The need to stay home and the impact of COVID-19 on essential workers and family. |
| T6  | positive, testing, tested, cases, patients, hospital, data | This topic focuses on data, particularly number of tests and cases. |
| T7  | masks, wear, face, hand, sanitizer, gloves, n95 | Things we can do to prevent COVID-19, e.g., masks and face coverings. |
| T8  | trump, china, americans, hoax, cdc, democrats, pandemic | US President Trump's statement of whether COVID-19 is a hoax and his discussion of China. |
| T9  | students, pandemic, nyc, petition, climate, college, university | A mix of discussion around school closures, the climate and the outbreak in New York City. |
| T10 | trump, house, white, president, press, vote, conference | USA politics including the white house press conferences and the US election. |
| T11 | test, cdc, facts, vaccine, fake, lab, control | Lab testing for COVID-19 and vaccination discussions, as well as discussing 'fake' tests available. |
| T12 | china, cases, death, wuhan, outbreak, spread, rate | Initial outbreak in Wuhan and it's associated case and death statistics. |
| T13 | time, old, years, day, feel, life, long | A mix of discussion around age and COVID-19 transmission, as well as time. |
| T14 | cases, deaths, york, million, state, total, cuomo | The statistics around deaths, particularly cases and deaths in New York City. |
| T15 | thanks, help, support, cruise, community, team, proud | Thank yous related to the pandemic mixed with discussion of cruise ship outbreaks. |
| T16 | health, need, care, public, emergency, fighting, stigma, curve, job | The need for health care related to addressing the COVID-19 pandemic in the USA. |
| T17 | money, pay, paper, toilet, buying, water, price | General economic concerns including pay and bulk buying. |
| T18 | stayhome, lockdown, order, quarantine, florida, beach, california | Quarantine and lockdown orders, particularly in Florida beaches and California. |
| T19 | shit, fucking, ass, wow, damn, dumb, hell | Negative reactions to COVID-19 and emotional usage of swearing. |
| T20 | break, trip, spring, school, classes, summer, quarnantinelife | COVID-19 school closures and spring break. |

Table A1: LDA generated topics and their interpretations.