# OpenReview forum: "Exploratory Analysis of COVID-19 Related Tweets in North America to Inform Public Health Institutes"
_EMNLP/2020/Workshop/NLP-COVID — NLP-COVID19-EMNLP Oral_

### Official Review · AnonReviewer2 · 2020-09-12
**Lacking novelty**

**Rating:** 4
**Confidence:** 5

**Review:**

**Recommendation:** Though mostly sound, the paper does not make a novel contribution; reject.

**Strengths**
1. Paper discusses non-US context
2. Researchers engage public-health researchers
3. Dataset and method are clearly described

**Weaknesses**
1. Approach is not novel - similar experiments have been performed on similar datasets
2. Proposed connection to potential interventions is not demonstrated
3. Not obvious how human-in-the-loop approach benefited anlaysis

### Review
The authors perform a topic analysis and sentiment analysis of an open Twitter dataset on the topic of COVID-19. The authors identify differences in the preference for topics between Canadian and American Twitter users. Additionally, they identify differences between sentiment across topics, including negative sentiment towards the "Asians".

Unfortunately, none of this is novel. As many as 14 papers have performed topic analysis COVID-19 Twitter data as of early May [Ordun et al., 2020](https://arxiv.org/abs/2005.03082). And indeed, even some of the paper's more nuanced findings, such as the implications of anti-Asian or sinophobic sentiment have been documented (e.g., [here](https://arxiv.org/pdf/2004.04046.pdf) and [here](https://arxiv.org/pdf/2005.12423.pdf))

Overall, the paper does is not novel enough in its approach, data, or findings to justify inclusion.

**Reproducability:** Researchers could easily reproduce this paper.

---

### Official Review · AnonReviewer1 · 2020-09-22
**The paper would benefit from a more fine-grained analysis**

**Rating:** 6
**Confidence:** 3

**Review:**

This work conducts experiments with topic modelling (using LDA) and aspect-based sentiment analysis (using a weakly-supervised approach) on a corpus of 320,000 tweets related to COVID-19 from Canada and the United States posted between January and May 2020. The findings show that there are both similarities and differences between the most popular topics for the Canadian and United States tweets. An analysis of topic changes over time reveals that topic popularity is similar between tweets from both countries, and that the popularity changes match with public health activities related to the specific topics (e.g., social distancing and the number of tests and cases). The sentiment analysis experiments show mixed sentiments for a few selected aspects.

The experiments for aspect-based sentiment analysis could be described in more detail. For instance, it is not specified how many public health experts edited the suggested lexicons and what the exclusion criteria for certain keywords were. It would also be helpful to provide a larger sample of identified aspects. Furthermore, the authors compute sentiments for 545 aspect terms, but only show results (Fig. 2) of a few selected aspects. Here, it would be interesting to see, for example, averaged results across aspects. It would also be interesting to analyze how the sentiment changed over time (similar to the topic modelling approach).

Overall, the proposed approach can be valuable for better understanding how people on social media react to the pandemic and corresponding governmental decisions. However, the analysis could be more fine-grained, and additional experiments would be beneficial to better demonstrate the potential of such an analysis.

---

### Official Review · AnonReviewer3 · 2020-09-25
**Topic modeling and aspect-based sentiment analysis of social media to inform public health decision making**

**Rating:** 5
**Confidence:** 5

**Review:**

The authors investigate twitter user’s reactions and concerns about COVID-19 in the US and Canada through topic modeling and aspect-based sentiment analysis. They report the change in the prevalence of 20 specific topics over time as well as inference of sentiment based on 545 aspect terms and 60 domain specific opinion terms.

While the described work is interesting, the authors did not evaluate their model and do not provide any evidence of the model accuracy. From the information provided in the paper, it is not possible to determine if the model does actually preform as expected and if the proposed approach can be reliably used to inform public health decision-making. While, theoretically, information obtained from sentiment analysis can be helpful in steering public health interventions, there is also a significant risk that false information can be detrimental by misguiding action and depleting limited public health resources. For this reason, evaluation of models that are used by health authorities requires rigorous validation before it can be considered for use.